# Strategy of Rainwater Discharge in Combined Sewage Intercepting Manhole Based on Water Quality Control

**Zhongqing Wei [1,2,\*], Xiangfeng Huang [1], Lijun Lu [1], Haidong Shangguan [2], Zhong Chen [3], Jiajun Zhan [3] and Gongduan Fan [3,\*]** 

[1] State Key Laboratory of Pollution Control and Resource Reuse, College of Environmental Science and Engineering, Tongji University, Shanghai 200092, China; hxf@tongji.edu.cn (X.H.); lulijun@tongji.edu.cn (L.L.)

[2] Fuzhou City Construction Design & Research Institute Co. Ltd., Fuzhou 350001, China; haidong.shangguan@fccdri.com

[3] College of Civil Engineering, Fuzhou University, Fujian 350116, China; N180520048@fzu.edu.cn (Z.C.); N180520053@fzu.edu.cn (J.Z.)

\* Correspondence: weizhongqing@fzwater.com (Z.W.); fgdfz@fzu.edu.cn (G.F.)

**Abstract:** In view of problems such as the poor control effect of combined sewage pollution caused by traditional intercepting weir and the limited extension of the urban drainage model, which needs a large amount of basic data, this paper not only studied the characteristics of mixed-flow pollution via the urban drainage model but also simulated and optimized 6 interception control strategies and proposed a water quality interception strategy based on the pollution concentration of combined sewage. The results showed that, compared with the traditional interception weir, the interception control strategy of rainwater discharge based on the mixed pipe network model can obviously improve the control rate of various pollutants and reduce the interception amount required for pollution control. Through optimization of the interception based on water quality control by the combination of chemical oxygen demand (COD) and $NH_4$-N, the interception rate was improved by 10.9% to 56.1% in contrast to the traditional interception weir and the closure water volume was reduced by 1432–6154 $m^3$, which effectively improved the reliability and economy of the interception.

**Keywords:** combined sewage pollution; combined sewage intercepting manhole; urban drainage model; water quality interception control

---

## 1. Introduction

The combined sewer system is a kind of drainage system in which domestic sewage and rainwater are mixed and drained in the same pipe culvert. In ancient times or in the early developing stages of the city, most drainage systems were the combined sewer system which has the characteristics of a low cost, simple pipeline and convenient construction. For example, in the 19th century, the domestic sewage of some countries in Europe were all discharged by the rainwater pipe network. However, with the continuous development of the city, the amount of the discharge of sewage is increasing day by day and the composition of the sewage is becoming more and more complex. Therefore, the pollution problem caused by the combined rainwater in the combined sewage system is becoming more and more serious. The pollution in combined sewage is one of the main factors that leads to black and odorous water in cities and it is prevalent in areas with rapid urbanization, dense population and the load of heavy surface pollution [1,2]. In addition to the overflow pollution in areas with combined sewerage system, the mixed flow of rainwater and sewage also exist in areas with separated drainage systems, which results from the mixed connection and the aging of the rainwater and sewage pipe

network. Many studies have found that the mixed discharge of rainwater and sewage could damage the water environment of the downstream river and cause black and odorous water. At the same time, emissions of pollutants and toxic substances contained in the combined sewage would threaten the health of residents [3,4]. In China, although all regions have fully recognized the negative impact of combined sewage pollution on the water environment, gray measures are still used to deal with this problem, which is not effective and needs difficult construction and a high engineering cost. Therefore, combined sewage pollution is still an urgent problem to be solved [5,6].

Interception of mixed rainwater is the most common control measure used in the transmission process or the end of the pipe network [7,8]. At present, many regions in China still adopt the traditional design of intercepting weir, which intercepts all sewage to the sewage treatment plant in sunny days and intercepts urban runoff and initial rainwater on rainy days. Besides, on rainy days, the combined sewage exceeding the designed closure rate of the intercepting manhole is discharged directly into the water body through the overflow pipe [9]. As we know, the interception ratio is a key parameter that affects the closure effect [10]. Practice has proved that the traditional intercepting manhole is difficult to control the pollution of mixed rainwater economically and efficiently [11,12]. This can be explained by the fact that the traditional interception weir is mostly based on the empirical and artificial water quantity calculation, while the interception of mixed rainwater possesses the characteristics of large regional difference, diverse pollutant sources and large concentration variations [8,13], which could lead to the overflow pollution when the intercepting amount is too small. Moreover, when the intercepting amount is too large, the traditional interception weir could result in low inflow concentration and the large flow rate of a sewage treatment plant, thus giving rise to the imbalance of sewage treatment operations. In order to solve the problem of combined rainwater pollution, the regulation of flow or water level is adopted in most practical projects and related research. Andrea Zimmer et al. studied the real-time hydraulic simulations coupled with control algorithms and explored a large number of potential changes to control procedures at short time intervals to provide dynamic feedback and optimization [14]. Meneses et al. investigated the combining traditional infrastructure solutions for urban drainage with real-time control (RTC) strategies and realized the reduction of the storage volume expansion otherwise needed to fulfill the environmental regulations for combined sewer overflow (CSO) discharge [15]. However, the practice shows that, to solve the problem of combined rainwater, more attention should be paid to the change of pollution concentration in rainwater and more urban drainage models are needed to realize the scheduling mechanism based on water quality and flow [16,17]. By using the United States Environmental Protection Agency's (USEPA) storm water management model (SWMM) for the hydraulic simulations, Rathnayake et al. proposed a new multi-objective evolutionary optimization method, which can realize the active control of intermittent dissatisfied discharged from combined sewer system [18]. Relevant studies have shown that the foundation of RTC based on the pollution process is available [19–21]. For example, the Lowell Regional Wastewater Utility had constructed and installed a comprehensive supervisory control and data acquisition (SCADA) system to manage and remotely operate the CSO stations based on real-time conditions, which are fairly sophisticated—typically including some flow screening, flow monitoring and CSO discharge pumping (when necessary) [22]. However, there are still rare studies or applications that use the RTC method based on the change of mixed rainwater pollutant concentration.

Therefore, in this study, Infoworks ICM (Infoworks ICM 7.5, Innovyze, UK) and ARCGIS (ARCGIS 10.2, Environmental Systems Research Institute Inc., Redlands, CA, USA) were used to build a mixed connection network model of a certain area in Fuzhou and the difference between the traditional interception scheme of mixed sewage and the water quality interception strategy based on pollutant concentration was compared. The water quality interception strategy based on COD and $NH_4$-N concentration was proposed to improve the intercepting efficiency. At the same time, the relationship between the pollution concentration characteristics of combined rainwater and the intercepting process control was described quantitatively, which can improve the reliability and economy of the

intercepting design and provide reference for the intercepting control technology based on changes in pollutant concentration.

## 2. Investigated Area and Methods

### 2.1. Regional General Situation

The area analyzed in this paper is located in the western suburb of Fuzhou, China, with a catchment area of about 0.217 km², a permanent population of about 40,000. Referring to 1985 national elevation benchmarks, the lowest and highest ground elevation is 6.0 m and 123.4 m, respectively (Figure 1). The land block type in this area is mainly residential land and the drainage system is the interceptive confluent system, however, the mixed connection of rain and sewage in this area is quite serious. In addition, in the study area, the drainage pipe network of DN600 and above is about 9.27 km and the drainage is collected in the main channel with a width of 3 m, a height of 2 m and a total length of about 1000 m through 5 branch pipes. The height of the inner bottom of the discharge outlet pipe at the end of the area is about 5.7 m from the ground.

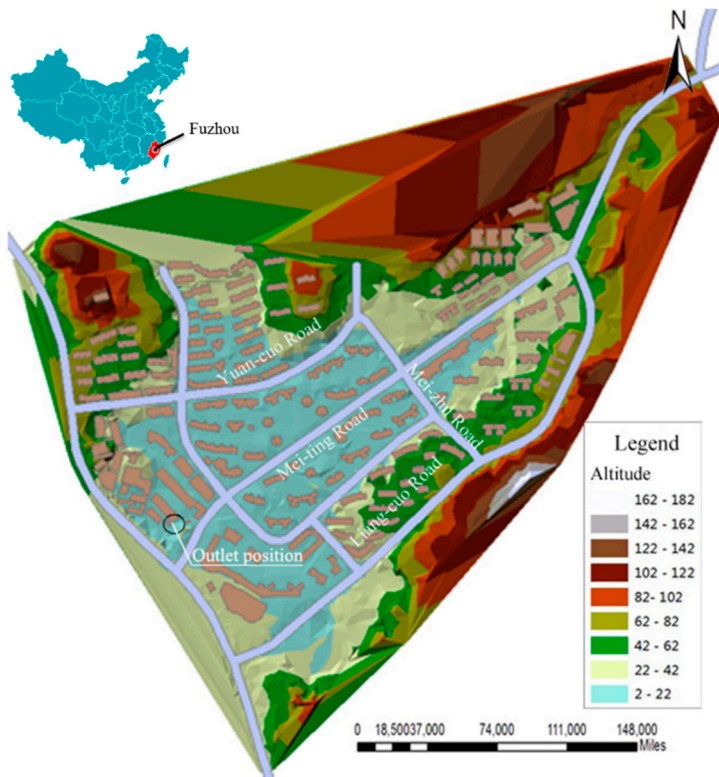

**Figure 1.** Regional pipe network and triangulated irregular network (TIN) model.

### 2.2. Acquisition of Basic Data

The professional teams of the pipe network were entrusted to carry out the pipe network survey, which included the survey of inspection wells, drainage pipes, drainage pump stations and other related facilities and parameters., All this research finally formed the topological relationship of the drainage pipe network.

The investigation of the current pollution situation included the mixed junction of the rainwater and sewage pipe network, the point of access to a non-point source pollution on dry days and the condition of the terminal storm drain. The volume method or flow-area method was used to determine the flow rate and the monitoring indexes of water quality included chemical oxygen demand (COD),

five-day biochemical oxygen demand ($BOD_5$), $NH_4$-N, total nitrogen (TN), total phosphorus (TP), suspended solid (SS) and pH value.

The rainfall data were monitored using the model SL3-1A rainfall monitoring alarm (Meteorological Instrument Factory co., LTD., Shanghai, China), which was installed on an open roof in the study area. The rainfall model used in the analysis was the Chicago rainfall model that fit well in the local area of Fuzhou and conformed to the local rainfall law. Moreover, the designed duration of rainfall is 2 h and the simulated return period corresponded to the rainfall events in years 1, 3 and 5, according to the expressions of rainstorm intensity for urban areas and county town (DBJ13-52-2003). The formula for the intensity of rainstorm as follow:

$$i = \frac{14.715(1 + 0.633 lgP)}{(t + 11.951)^{0.724}} \tag{1}$$

where *i* is the rainfall intensity, mm/min; *P* is the design frequency, *a*; and *t* is the duration of rainfall, min.

## 3. Model Construction and Calibration

### 3.1. Hydraulic Model Construction and Check

The construction of the hydraulic model mainly includes the construction of a drainage system, the division of the catchment area and the determination of runoff parameters. The main data imported by the drainage system contain the node number, X, Y, Z and the node number of upstream and downstream of the pipe and canal, the bottom elevation of upstream and downstream and the width and height of the pipe and canal. The TIN model is generated by using 3D analyst tools' data management of ArcGIS and the conversion tool converts the generated TIN into the DEM model to form the ridge line on both sides of the region and the catchment boundary. The catchment boundary and ridge line were used to determine the boundary of the study area and the Thiessen polygon was used to divide subset water area expect for the mountain in the study area.

The underlying surface of this area is mainly divided into roof, road surface, green space and mountain. The runoff model, including road surface, roof and mountain, adopts a fixed runoff coefficient method. The Horton infiltration model is adopted for green land and the SWMM model is adopted for all confluence models. In this paper, a representative rainfall event with long duration and heavy rainfall (9 August 2015) was selected for the model's 2-d hydraulic model verification. The rainfall lasted for 2040 min, with a total rainfall of 279.5 mm and a maximum rainfall intensity of 60 mm/h. For the parameters of confluence and production, the reasonable value range was first collected and then the parameters were repeatedly adjusted based on the principle which stuck to the difference between the simulated value and the measured value in the water point distribution and water depth should be the minimum. The final debugging results showed that the maximum depth of water in the water section of Meiting Road reached 50–60 cm and the depth of water in the middle section of Yuancuo Road reached 80–90 cm. The simulation results were consistent with the actual survey results, indicating that the established model was accurate and reliable. The parameters of the final confluence model are shown in Table 1.

**Table 1.** Parameters of the production confluence model.

| Underlying Surface | Runoff Yield Model | Runoff Coefficient | Initial Loss (mm) | First Infiltration Rate (mm/h) | Final Infiltration Rate (mm/h) | Attenuation Rate (mm/h) | Confluence Model | Confluence Parameter |
|---|---|---|---|---|---|---|---|---|
| Roof | Fixed | 0.95 | 3 | - | - | - | SWMM | 0.014 |
| Road | Fixed | 0.9 | 3 | - | - | - | SWMM | 0.014 |
| Green land | Horton | - | 12 | 79.38 | 13.42 | 4.34 | SWMM | 0.3 |
| Mountain | Fixed | 0.6 | 12 | - | - | - | SWMM | 0.4 |

### 3.2. Simulation and Fitting of Dry Weather

According to the investigation, there were 38 main mixed connections of rainwater and sewage. The water quality and quantity of the four main mixed connections and the end discharge outlets were monitored and the monitoring time was from 8:00 to 20:00, lasting for 5 days.

In accordance with the distribution of the urban population, the obtained 5 monitoring water quantity data and the field investigation, the daily flow rate of dry season sewage at the end the discharge outlet was determined. In addition, the discharge outlet flow was distributed to the upstream mixed junction based on the analysis results and the most unfavorable inflow situation was considered at the same time. The fitting results of the output flow are shown in Figure 2.

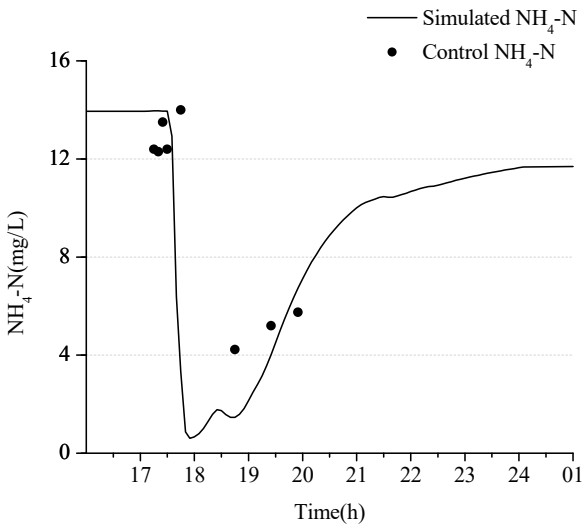

**Figure 2.** Fitting diagram of sewage flow in dry season.

The main fitting water quality indexes of dry sewage are SS, COD, TN, TP and $NH_4$-N. In this paper, the measured rainfall time was in the afternoon and notably, the value of TN, $NH_4$-N in the morning and afternoon differed over 200%. Therefore, the measured mean value in the afternoon was adopted and the measured mean value in the whole day was adopted for other indexes. The simulated value and control value are shown in Table 2. The reason why the value C is significantly higher in comparison with other SS is that the combined rainwater in this area is very heavy and the effluent is greatly affected by the surrounding mixed sewage. Therefore, due to the existence of restaurants and car washes around the test site, there will be a large amount of wastewater flowing into the pipe from 12:00 to 2:00, resulting in an obvious increase of SS value.

**Table 2.** Fitting table of sewage quality in dry days.

| Value | Test Time (h: min) | SS (mg/L) | COD (mg/L) | TN (mg/L) | TP (mg/L) | $NH_4$-N (mg/L) |
|---|---|---|---|---|---|---|
| Average value A | 9:00 | 31 | 73.9 | 45.1 | 2.29 | 43.1 |
| Average value B | 12:00 | 23 | 82.6 | 44.2 | 1.95 | 42.3 |
| Average value C | 14:00 | 206 | 91.5 | 22.1 | 1.85 | 18 |
| Average value D | 20:00 | 92 | 92.6 | 22.4 | 1.83 | 15.4 |
| Average value | / | 88 | 85.2 | 22.25 | 1.98 | 16.7 |
| Simulation values | / | 86.3 | 83.9 | 22.3 | 1.91 | 16.8 |
| △ (%) | / | −1.93 | −1.47 | 0.22 | −3.54 | 0 |

After simulation, the variation range of water quality simulation value and control value was within 4% and the quantity of sewage in dry season met the actual control requirements.

*3.3. Water Quality Checking on Rainy Days*

Combined sewage discharged on rainy days mainly includes surface runoff, sewage and sewage sediment [23]. The parameters in the model mainly involve three processes which are surface sediment and the pollutant accumulation process, scouring process and its cumulative scouring process in the pipeline. The catchment area is divided into green area, residential area (low density), residential area (high density) and commercial area. Surface sediment accumulation factor Ps (as shown in Table 3), pollutant efficiency factor Kpn and corresponding water quality parameters of each functional area were used to define the surface accumulation process. Desbordes model was selected to simulate the surface erosion process and the ackers white model was used to simulate the cumulative erosion process of pipeline sediments and contaminants. The rainfall of the model was based on the measured rainfall data (total rainfall of 4300 m$^3$) of 17:05 on 28 July 2016 obtained by the regional rain gauge. The rainy day verification mainly included the adjustment of catchment area, parameter sensitivity analysis and the verification of the measured and simulated values at the end discharge outlet. The simulated curves and measured values of various pollution indicators are shown in Figure 3. The relative error between the simulated values and measured values was within 15% and the model met the requirements of engineering analysis.

**Table 3.** Reference value of sediment accumulation rate [24].

| Land Type | $P_s$ (kg/km$^2$/day) |
|---|---|
| Residential area (high density) | 0.25 |
| Residential area (low density) | 0.06 |
| Central City (commercial Area) | 0.25 |
| Green area | 0.06 |

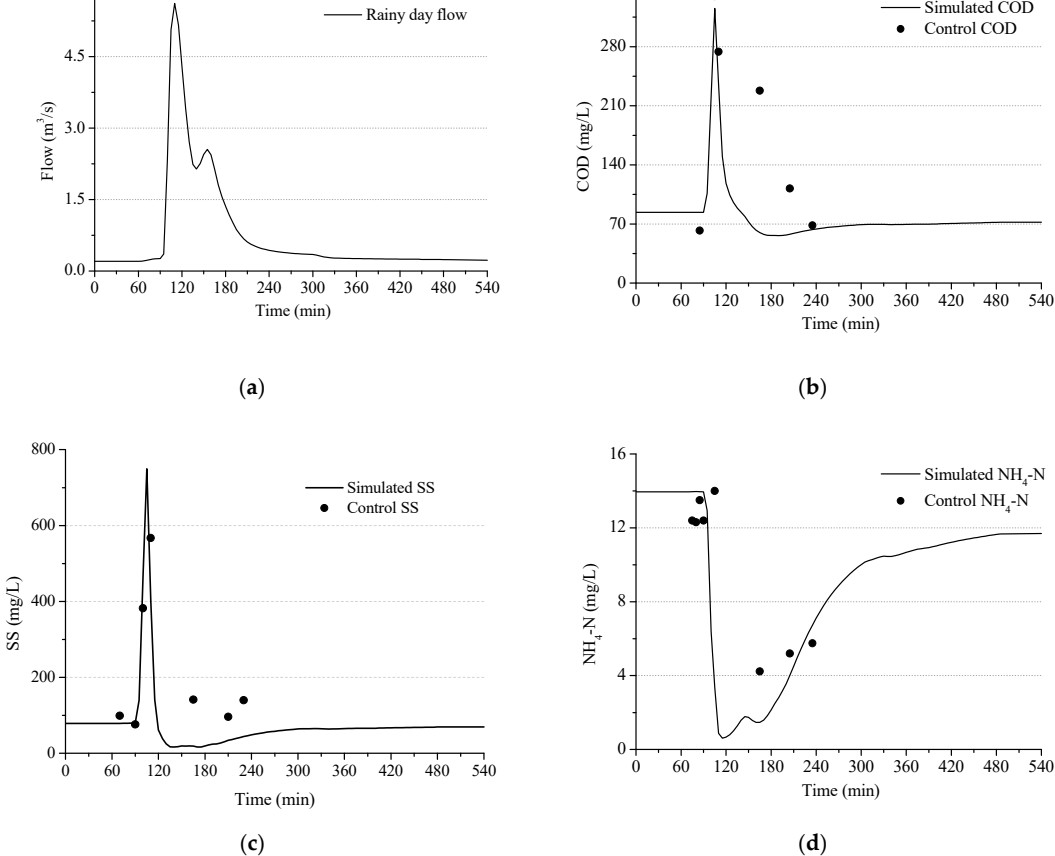

**Figure 3.** *Cont.*

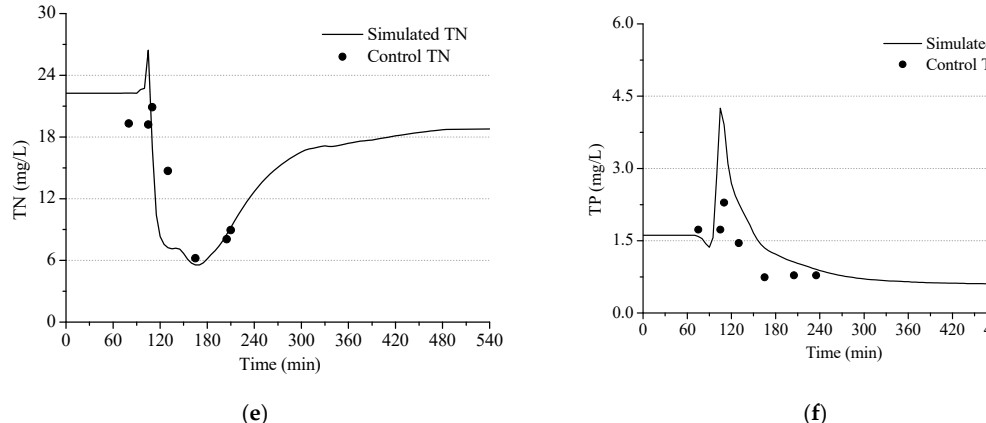

(**e**)　　　　　　　　　　　　　　　　　　　　　　　　(**f**)

**Figure 3.** Simulation curves and measured values of each pollution index, (**a**) flow, (**b**) chemical oxygen demand (COD), (**c**) suspended solid (SS), (**d**) ammonia nitrogen (NH$_4$-N), (**e**) total nitrogen (TN), (**f**) total phosphorus (TP).

## 4. The Construction and Calibration of the Model

### 4.1. Characteristics of Rainwater Drains Pollution

#### 4.1.1. The Measured Rainfall

As can be seen from Figure 3, mixed-flow rainwater was generated after 20 min rainfall and the flow rapidly increased to 5.60 m$^3$/s after 15 min discharge. Besides, accompanied by the initial rainfall, the concentrations of SS, COD and TP increased sharply to 745.3 mg/L, 325.6 mg/L and 4.2 mg/L, which were 9.4, 3.9 and 2 times higher than that in the dry season, respectively. All above showed a typical phenomenon of an initial scour of urban sewage pipes, which exhibited the characteristics of a large initial instantaneous flow, strong impact and high pollution load [25,26].

The correlation of pollution indexes of mixed rainwater is shown in Table 4. According to the table, the discharge flow of mixed rainwater was positively correlated with the pollution concentration of COD, TP and SS and negatively correlated with TN and NH$_4$-N. Meanwhile, Pearson correlation coefficients were all greater than 0.4, indicating that the correlation of various pollutants in mixed rainwater flow was above medium.

**Table 4.** Correlation analysis of pollution indexes.

| Pearson Correlation Coefficient | Flow Rate | COD | TN | TP | NH$_4$-N | SS |
|:---:|:---:|:---:|:---:|:---:|:---:|:---:|
| Flow | 1 | 0.671 | −0.454 | 0.855 | −0.786 | 0.449 |
| COD | 0.671 | 1 | 0.268 | 0.941 | −0.22 | 0.95 |
| TN | 0.454 | 0.268 | 1 | 0.011 | 0.881 | 0.433 |
| TP | 0.855 | 0.941 | 0.011 | 1 | −0.448 | 0.791 |
| NH$_4$-N | 0.786 | −0.22 | 0.881 | −0.448 | 1 | −0.032 |
| SS | 0.449 | 0.95 | 0.433 | 0.791 | −0.032 | 1 |

#### 4.1.2. The Designed Rainfall

The discharge curve of the end discharge outlet after simulation is shown in Figure 4. The maximum flow rate of mixed rainwater with the return period of 1, 3 and 5 years was 15.67 m$^3$/s, 18.66 m$^3$/s and 19.61 m$^3$/s, respectively. From the perspective of the discharge flow curve, the peak flow of the end discharge and the duration were positively correlated with the rainfall intensity, which was consistent with the study of Sandoval et al. (2013) on CSO in Berlin [18]. At the same time, the amount of pollutants discharged with different recurrence periods was assessed (as shown in Table 5) and it was found that the mixed flow was heavily polluted in this region, which must be controlled by effective measures.

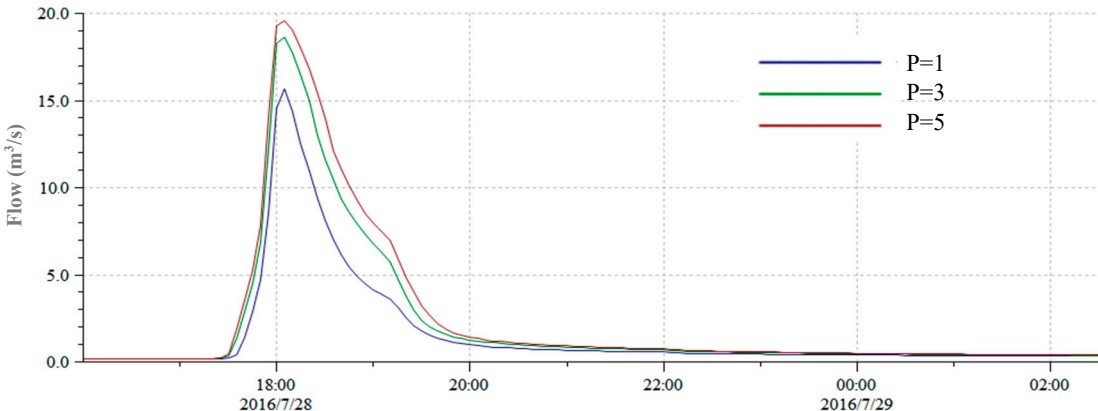

**Figure 4.** Discharge curves under different recurrence periods.

**Table 5.** Total flow and pollutant discharge of intercepting manholes under different recurrence intervals.

| P (a) | Total Flow (m³) | SS (kg) | COD (kg) | TN (kg) | TP (kg) | NH₄-N (kg) |
|---|---|---|---|---|---|---|
| 1 | 59416 | 2894.88 | 2298.20 | 204.45 | 55.30 | 76.24 |
| 3 | 80169 | 4669.86 | 2271.32 | 307.55 | 54.87 | 102.81 |
| 5 | 91256 | 4667.85 | 2278.26 | 305.88 | 54.99 | 155.51 |

*4.2. Intercepting Simulation of Traditional Interception Weir*

The interception simulation of the traditional interception weir mainly includes the simulation of interception weir, interception pipe and detention tank (as shown in Figure 5). In order to ensure the intercepting effect, the interception ratio $n = 5$ was adopted. The closure pipe diameter was DN600 and the effective volume of the detention tank was 30,000 m³. In dry days, sewage in rainwater pipes is transported to the sewage treatment plant through intercepting pipes, while on rainy days, part of the combined rainwater flows into the detention tank.

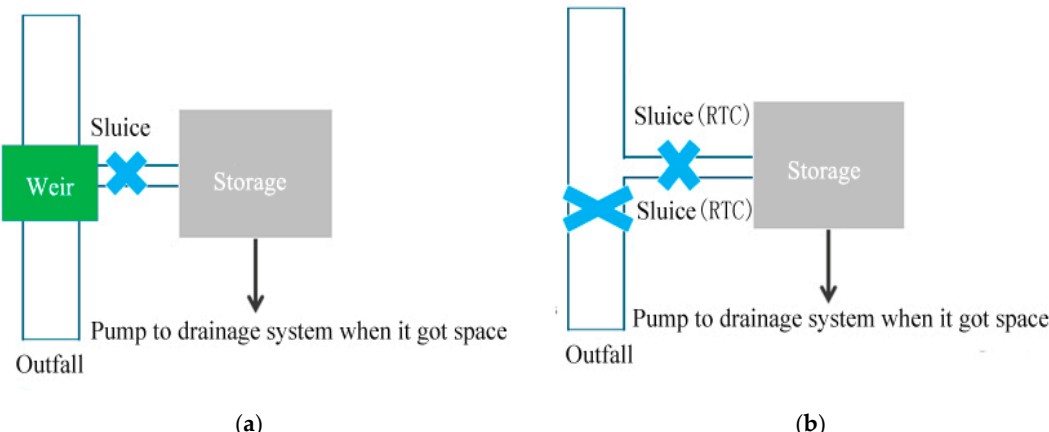

**(a)**                                        **(b)**

**Figure 5.** (**a**) Schematic diagram of simulation of traditional interception strategy; (**b**) Water quality interception strategy based on the pollution concentration of mixed rainwater.

According to Figure 6, the trend of the interception flow curve was similar to that of the terminal discharge outlet confluence curve and the water volume reached the maximum around 18:05. In addition, with the increase of rainfall intensity, the closure volume increased correspondingly.

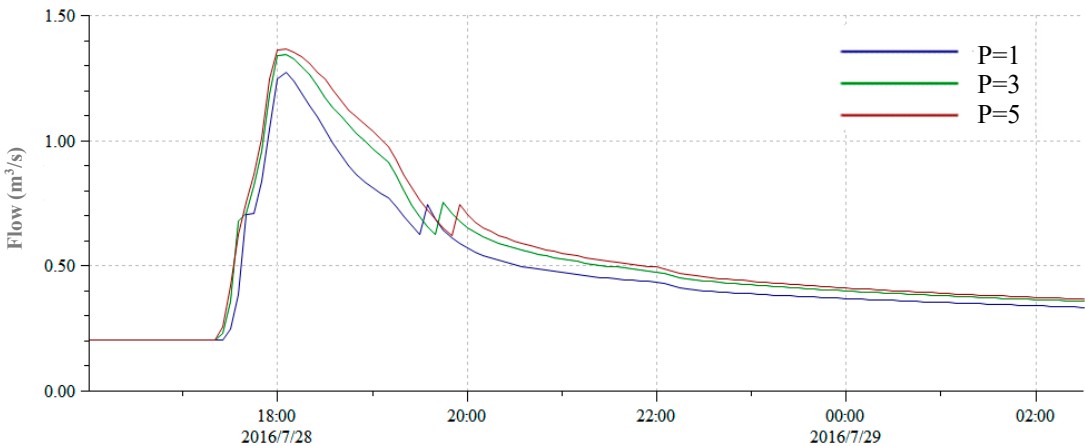

**Figure 6.** Interception flow curve of traditional interception weir.

As can be seen from Table 6, the maximum interception rate of SS, COD and TP of the traditional interception weir was only 34% and the intercepting flow of these three pollutants was low. Therefore, in comparison, the intercepting flow of NH$_4$-N and TN was relatively high. When the recurrence interval was 1 year, the interception rate of NH$_4$-N and TN were 62% and 93%, respectively. While with the increase of rainfall intensity, the interception rate of NH$_4$-N and TN were respectively reduced to 36% and 39% with the return period of 5 years and were inefficient and unstable. In consequence, the traditional interception weir was difficult to control all kinds of pollutants stably and it was necessary to explore more closure measures in line with the local characteristics of pollution [11].

**Table 6.** Interception statistics of traditional interception weir.

| P (a) | Flow Interception (m³)/(%) | SS Interception (m³)/(%) | COD Interception (m³)/(%) | TP Interception (m³)/(%) | TN Interception (m³)/(%) | NH$_4$-N Interception (m³)/(%) |
|---|---|---|---|---|---|---|
| 1 | 16922/33 | 920.27/32 | 636.17/28 | 18.77/34 | 126.62/62 | 71.18/93 |
| 3 | 18790/27 | 874.53/19 | 580.85/26 | 12.51/23 | 114.90/37 | 64.42/63 |
| 5 | 19564/25 | 860.08/18 | 560.21/25 | 12.11 /22 | 109.65/36 | 61.16/39 |

### 4.3. The Simulation of Water Quality Interception

#### 4.3.1. Water Quality Interception Strategy

Through the simulation, it is found that various pollutants would show different pollution characteristics with the change of discharge flow, however, the traditional intercepting design of the interception weir ignores this point and focuses more on the interception control of water volume, thus resulting in the unsatisfactory closure effect [27]. Based on previous data, mixed rainwater pollution and discharge flow were moderately correlated. Meanwhile, after the initial sharp increase in pollution concentration, the concentration decreased rapidly and remained relatively stable. Moreover, the polluted concentration of mixed-flow rainwater was lower than that of sewage in dry weather, which provided a practical basis for the RTC (real-time control) technology based on water quality closure in this region. Therefore, considering the effect of closure and the practicability of measures, this paper proposed a water quality interception strategy based on the pollution concentration of mixed-flow rainwater, which could dispatch the gate through RTC technology (as shown in Figure 5).

In general, this strategy took the water quality of combined rainwater as the control parameter of RTC. When the water quality index of combined rainwater was greater than or equal to the water

quality limit stipulated by the national standard, the combined rainwater was intercepted through the intercepting pipe and retained in the detention tank. The specific control conditions are as follows:

$$\frac{C_{dr}Q_{dr}+C_RQ_R}{Q_{dr}+Q_R} \geq C_s \tag{2}$$

where $C_{dr}$, $C_R$ represents the concentration of pollution from drought and rainwater, mg/L; $Q_{dr}$ is the Pollution flow in dry weather, $m^3/s$; $Q_R$ is the flow, $m^3/s$; and $C_S$ is the discharge standard of mixed flow of rainwater, mg/L.

The equation is a control mode for overflow pollution with the combination of water quality and flow, which can not only ensure that the overflow sewage pollutants on rainy days meet the required discharge standard but also provide the basis for the real-time online scheduling. First grade A of China's urban sewage discharge standard (GB18918-2002) is selected as the standard in this simulation and the control concentration is shown in Table 7.

**Table 7.** Standard values of first grade A sewage discharge.

| Index | SS | COD | TP | TN | NH$_4$-N |
|---|---|---|---|---|---|
| Value(mg/L) | 10 | 50 | 0.5 | 15 | 5 |

### 4.3.2. The Simulation and Comparison of Water Quality Closure

In order to verify the feasibility of the water quality interception strategy based on the pollution concentration, five interception strategies based on SS, COD, TN, TP and NH$_4$-N were simulated. When the instantaneous sewage concentration was higher than or equal to the standard concentration of grade A, the sluice gate would be opened and the sewage would enter the detention tank. When the instantaneous sewage concentration was lower than the standard value of grade A, the sluice gate of the intercepting pipe would be closed and the combined rainwater would be discharged into the receiving water body. The statistics of intercepting results of various water qualities are shown in Table 8.

**Table 8.** The control simulation of various water quality closures.

| Index | Flow Interception (m$^3$)/(%) | SS Interception (m$^3$)/(%) | COD Interception (m$^3$)/(%) | TP Interception (m$^3$)/(%) | TN Interception (m$^3$)/(%) | NH$_4$-N Interception (m$^3$)/(%) |
|---|---|---|---|---|---|---|
| | | | P = 1 | | | |
| SS | 20786/35 | 2599.71/90 | 1546.71/67 | 36.39/66 | 168.95/83 | 66.63/87 |
| TP | 31388/53 | 2717.31/94 | 1945.87/85 | 47.01/85 | 175.02/86 | 61.79/81 |
| TN | 14744/29 | 521.98/18 | 540.18/24 | 10.67/19 | 98.30/48 | 64.16/84 |
| NH$_4$-N | 9019/18 | 533.62/18 | 469.65/20 | 9.49/17 | 81.34/40 | 54.61/72 |
| COD | 6904/12 | 2257/30 | 1054.71/46 | 26.49/48 | 88.31/43 | 11.55/15 |
| | | | P = 3 | | | |
| SS | 28900/36 | 3680.71/79 | 1502.51/66 | 35.49 /65 | 228.59/74 | 86.32/84 |
| TP | 30695/38 | 4221.17/90 | 1766.76/78 | 42.48 /77 | 229.26/75 | 66.65/65 |
| TN | 17695/26 | 589.03/13 | 505.10/22 | 10.02/18 | 121.36/39 | 77.90/76 |
| NH$_4$-N | 7557/11 | 342.57/7 | 307.93/14 | 6.08/11 | 73.28/24 | 47.93/47 |
| COD | 10082/13 | 2868.24/61 | 974.97/43 | 24.47/45 | 115.33/37 | 18.93/18 |
| | | | P = 5 | | | |
| SS | 24985/27 | 3590.82/77 | 1476.84/65 | 34.82 /63 | 223.76/73 | 125.04/80 |
| TP | 28976/32 | 4123.82/88 | 1683.06/74 | 41.07 /75 | 216.08/71 | 92.39/59 |
| TN | 20039/25 | 509.04/11 | 476.33/21 | 9.32/17 | 117.59/38 | 99.89/64 |
| NH$_4$-N | 4717/6 | 263.49/6 | 215.60/9 | 4.31/8 | 50.66/17 | 39.69/26 |
| COD | 9591/11 | 2779.16/60 | 946.59/42 | 23.73/43 | 110.65/36 | 35.46/23 |

Compared with the above results of water quality interception, the average closure rate of various pollutants from high to low was TP > SS > COD > TN> NH$_4$-N and most of them were inversely proportional to rainfall intensity. This is because mixed flow rainwater is less affected by non-point source pollution at low rainfall intensity. In addition, the average interception rate of pollutants based on the strategy which focused on the interception of SS and TP was more than 71.7%, while the average interception rate of traditional interception weir was only 28.0% to 49.7%. From the above, these two interception methods have significantly improved the interception effect of pollutants.

The average interception rate of pollutants based on COD interception was 40.6% to 46.0%, which was because the COD would reach the standard of level I A at 17:59 after the rainfall. The COD concentration of mixed rainwater in the later period was all lower than 50 mg/L (as shown in Figure 7a) and the subsequent pollutants were not intercepted. Besides, there is little difference between TN intercepting and traditional intercepting weir. Statistics showed that the average intercepting rate of pollutants based on NH$_4$-N intercepting was only 13.0% to 33.4%, which was lower than the control of traditional intercepting weir. This could be explained by the reason that NH$_4$-N was lower than 5 mg/L after 17:38 rainfall (as shown in Figure 7b) and the closure control ended prematurely with the lowest water volume, which was only 897 m$^3$.

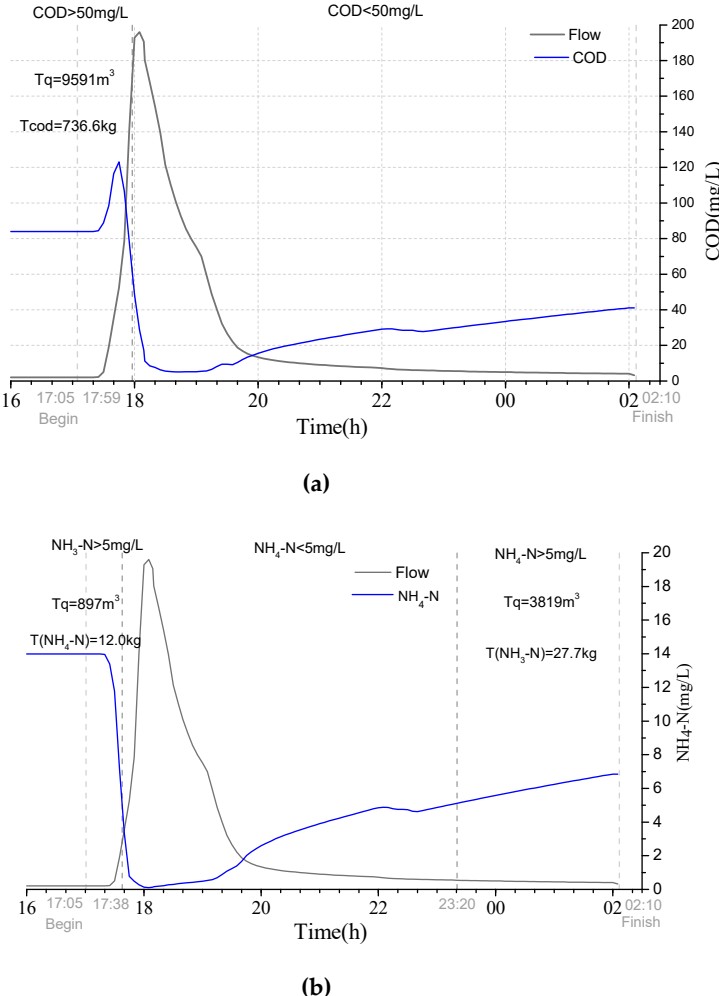

**Figure 7.** Process of COD (**a**) and NH$_4$-N (**b**) pollution with return period of 5 years.

*4.4. The Optimization of Interception Strategy*

According to the comparison of various interception methods, we can see that the closure effect based on SS and TP was the best, however, the construction cost of these two methods were expensive

due to the large flow of the intercepting sewage. In addition, the low pollution concentration of the interception based on COD in the later period would lead to no interception in the later stage and the earlier period of the interception based on $NH_4$-N was too short. Thus, considering that the correlation between SS, TP and COD was above 0.9, COD + $NH_4$-N can be combined to intercept the flow from the perspective of the intercepting control, that is, the control index of COD was adopted before the rainfall peak and the control index of $NH_4$-N was adopted at the later period of the rainfall peak. The intercepting statistics are shown in Table 9.

**Table 9.** Combined results of COD + $NH_4$-N.

| P (a) | Flow Interception (m$^3$)/(%) | SS Interception (m$^3$)/(%) | COD Interception (m$^3$)/(%) | TP Interception (m$^3$)/(%) | TN Interception (m$^3$)/(%) | $NH_4$-N Interception (m$^3$)/(%) |
|---|---|---|---|---|---|---|
| 1 | 15490/30 | 2726.95/94 | 2016.35 /88 | 45.88/83 | 169.65/83 | 55.49/73 |
| 3 | 16730/24 | 3176.18/68 | 1810.44/80 | 41.57 /76 | 188.60/61 | 56.92/55 |
| 5 | 13410/17 | 2994.76/64 | 1692.44/74 | 39.13 /71 | 161.30/53 | 63.13/41 |

According to Table 9, the average intercepting rate of all pollutants based on the COD + $NH_4$-N interception strategy was 60.6%−84.1%, which was about 10.9%−56.1% higher than the traditional interception strategy and significantly improved the closure effect. Meanwhile, the closure water volume of the COD + $NH_4$-N interception was lower than that of the traditional interception weir and the decreased amount was about 1432−6154 m$^3$, which can effectively reduce the closure construction cost. Moreover, as shown in Figure 8, compared with the traditional interception weir, the closure water volume with the interception strategy based on the combination of COD + $NH_4$-N decreased by 7.82% and the closure rate of this combined interception strategy to all pollutants increased, especially the SS, COD and TP which respectively increased by 45.73%, 49.70% and 49.15%, resulting in the significant improvement of the intercepting efficiency. In summary, the combined COD + $NH_4$-N intercepting strategy was better than the traditional intercepting weir in both control effect and economic efficiency.

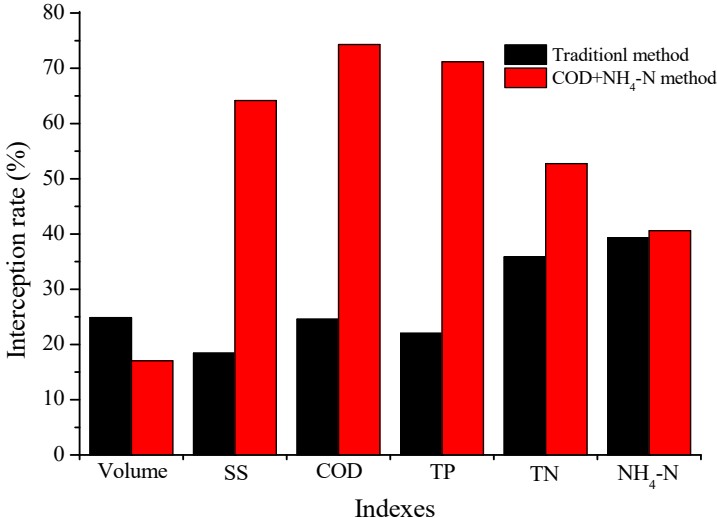

**Figure 8.** Comparison of interception strategy before and after optimization with the return period of five years.

## 5. Conclusions

In this paper, a control strategy based on water quality interception was proposed. This strategy, through the RTC of the discharge and the concentration of various pollutants, could not only reduce the combined overflow pollution but also effectively optimize the reliability and economy of the interception facility. The main conclusions were as follows:

(1) By the simulation of intercepting control of combined sewage, it was found that the effect of the traditional intercepting weir on the overflow pollution is limited. Meanwhile, in the combined rainwater, the correlations between various pollutants and the flow were all medium or above. Besides, the intercepting sluice gate was scheduled through the simulation of the RTC and the overflow pollution control equation combining the concentration and the flow was proposed, which ensured that the discharge of the combined rainwater met the first grade A of China's urban sewage discharge standard.

(2) Through the simulation and comparison of five kinds of water quality interception control, the best water quality interception control strategy was put forward, which was based on the combination of COD and $NH_4$-N. When the recurrence interval was 1, 3 and 5 year, the interception volume was decreased by 1432 $m^3$ (8.5%), 2060 $m^3$ (10.9%) and 6154 $m^3$ (31.5%), respectively. The amount of the overflow pollution of various pollutants (SS, COD, TP, TN, $NH_4$-N) was decreased by 34.4%, 34.6% and 32.5%, respectively. All above indicated that the efficiency of the interception was greatly improved and the cost in the construction and operation of the traditional interception facilities could be reduced. In addition, through the consideration of the application method in this intercepting strategy, it is possible to apply this strategy to other cases. This strategy provided a control mode for the overflow pollution existed in different areas. Furthermore, the construction and application of the model, the RTC (real-time control) technology based on water quality interception and the method used for the optimization of interception strategy proposed in this paper all have a useful reference value for the optimization of application in different cases.

(3) With further development of the parameter measurement technology of flow and water quality, the dynamic drainage model based on the water quality is helpful to the quantitative analysis of combined rainwater pollution, which can make up for the deficiency of the traditional interception weir. Furthermore, the dynamic drainage model can also provide a better understanding of the combined overflow pollution process and formulate the control strategy for the control of the combined rainwater pollution according to local conditions.

**Author Contributions:** Z.W. supported programming and software application, reviewed statistical models and model development, and provided significant suggestions on the model results and management scenarios. The manuscript was completed under the supervision by X.H. and L.L.; H.S. helped define the statistical model, produced figures, and led the drafting of the paper; G.F. supported the image of data and provided helpful insights into physical processes and data analysis. G.F., J.Z. and Z.C. wrote the draft checked the grammar and spelling mistakes in the manuscript. All authors have read and approved the final manuscript.

**Funding:** This research was financially supported by Science and Technology Project Program of Fujian (No. 2019Y3003), National Natural Science Foundation of China (No. 51778146) and the Outstanding Youth Fund of Fujian Province in China (No. 2018J06013).

**Acknowledgments:** The authors would like to thank the reviewers and editors for their valuable remarks and comments that greatly improved the quality of the paper.

**Conflicts of Interest:** The authors declare no conflict of interest.

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
