# Peer review of "Strategy of Rainwater Discharge in Combined Sewage Intercepting Manhole Based on Water Quality Control"

_water, doi:10.3390/w11050898_

Round 1

Reviewer 1 Report

It seems like introduction part is not well documented and requires improvements in terms of previous research, models and methods.

line 69. CSO stations?

line 74. Infoworks ICM and ARCGIS, Please mention the software version and vendor details in reference. which version of ArcGIS is used.

furthermore, directly writing the abbreviations. such as COD, SS, TP, Its better to write the complete name first time then you can use them.

line 86. I suggest you to write the area in kilometer-squares instead of ha.

line 87. to line 92. The details are not coherent and its very difficult to establish the link between provided information and corresponding map. elevation values are not written professionally, 6.0 meter to 123 meter. 

line. 93. map should be improved by putting the legend for DEM elevation. nodes should be displayed on the map so reader can establish the link between map and text. Furthermore, DEM looks very strange around mountains, it looks like a TIN model rather than a DEM. location map (a) can be reduced as inset map and study area map (b) can be enlarged (two times is better).

line 106. please look for your spellings

line. 108 intensity formula is not well explained and there is no reference for this formula. do you own this formula?

line 118. need to mentioned which ArcGIS software version you are using and please refer to ESRI tutorials and software manual to clearly mention the name of the tool and extension you are using for this work, for instance, conversion tool, which conversion tool?,

line 119. show the catchment boundary and ridge line in Figure 1. The subset water area within the catchment boundary is divided by Thiessen polygon? does not make sense, what are you trying to say?

at line 106. you mentioned rainfall duration was 2 hours, however, at line 125 rainfall lasted for 2040 mins, 

153. Table 2. SS average value C is significantly higher as compared to other readings of SS. Why?, you do not explain

line 161, 162. please describe the models you have mentioned, better idea is to include them in literature review so you can site them in methodology. 

line. 165. adjustment of catchment area? As I know you already defined your catchment boundary. how and why you need to adjust your catchment boundary.

line 176. you refer to figure 3. In text you have mentioned the time in minutes, however, the graph has been designed in hours, its difficult to correlate, please improve the figures and be consistent with units used in text and in figures.

line 241. equation is not explained and cited 

line 305. Conclusion is not enough, requires further improvements.

line 305. RTC?

Author Response

Thanks a lot for your attention paid to this work. We do appreciate the professional and illuminating comments provided by the reviewers. These advices and suggestions are valuable for both this work and our subsequent research. We have tried our best to revise the manuscript (Manuscript ID water-489896) based on the comments. We hope the revision can be satisfactory. All changes made in the revised manuscript are highlighted in red font. The illustration of the revisions is shown point by point as follows.

Finally, thank you very much for your comments and suggestions again.

Best regards,

Gongduan Fan, Zhongqing Wei

2019-4-20

It seems like introduction part is not well documented and requires improvements in terms of previous research, models and methods.

Response: Thanks for the valuable comment. The authors have improved the introduction, and the changes are highlighted in red font in the part of the introduction.

line 69. CSO stations?

Response: Thanks for your carefulness. “CSO” is the abbreviation of “combined-sewer overflow”. The CSO stations is part of the sewer collection system to regulate the flow along the interceptor. The CSO stations are fairly sophisticated-typically including some flow screening, flow monitoring, and CSO discharge pumping (when necessary). The authors have added the complete name where the word first appeared. It can be found in Line 74, Page 2.

line 74. Infoworks ICM and ARCGIS, please mention the software version and vendor details in reference. which version of ArcGIS is used.

Response: Thank for your comment. The authors have added the version and vendor of these software to the paper. These changes can be found in Line 88 and Line 89, Page 2.

furthermore, directly writing the abbreviations. such as COD, SS, TP, it’s better to write the complete name first time then you can use them.

Response: Thanks for your valuable comment. The authors have added the complete name where the word first appeared and the changes can be found in Line 120, Page 4, Line 121 and Line 122, Page 4.

line 86. I suggest you to write the area in kilometer-squares instead of ha.

Response: Thanks for your valuable comment. The authors have used kilometer-squares instead of ha and the change can be found in Line 101, Page 2. The same problem in table 3 has also been modified.

line 87. to line 92. The details are not coherent and its very difficult to establish the link between provided information and corresponding map. elevation values are not written professionally, 6.0 meter to 123 meters.

Response: Thanks for your valuable comment. The authors have modified the Figure 1 to make it easier to be understood and revised the expression of elevation values. The changes can be found in Line 101, Page 3.

line. 93. map should be improved by putting the legend for DEM elevation. nodes should be displayed on the map so reader can establish the link between map and text. Furthermore, DEM looks very strange around mountains, it looks like a TIN model rather than a DEM. location map (a) can be reduced as inset map and study area map (b) can be enlarged (two times is better).

Response: Thanks for your valuable comment. The authors have put the legend for elevation and displayed the nodes in the map. The figure has been modified in strict accordance with the opinions of reviewers. The authors agree that it looks like a TIN model rather than a DEM model. The authors intended to combine the TIN model and DEM elevation map to show the general situation of the area more clearly. In fact, it is the TIN model and the authors have modified the figure caption.

line 106. please look for your spellings

Response: Thanks for the comment. Sorry for our carelessness. It has been revised and can be found in Line 126, Page 4.

line. 108 intensity formula is not well explained and there is no reference for this formula. do you own this formula?

Response: Thanks for the comment. The intensity formula is the designed rain pattern formed according to the expressions of rainstorm intensity for urban areas and county town (DBJ13-52-2003). The authors have added the explanation and the reference in Line 128, Page 4.

line 118. need to mentioned which ArcGIS software version you are using and please refer to ESRI tutorials and software manual to clearly mention the name of the tool and extension you are using for this work, for instance, conversion tool, which conversion tool?

Response: Thanks for your comment. The authors have added the version and vendor of ArcGIS, and the conversion tool is 3D Analyst of ArcGIS.  The change can be found in Line 89, Page 2.

line 119. show the catchment boundary and ridge line in Figure 1. The subset water area within the catchment boundary is divided by Thiessen polygon? does not make sense, what are you trying to say?

Response: Thanks for your carefulness. The catchment boundary and ridgeline in Figure 1 were used to determine the boundary of the study area, while Thiessen polygon was used to divide the subset water area in the study area. In addition, as for the division of subset water area, the authors compared three different methods, including Thiessen Polygon partition, GIS partition and manual partition. The authors found that the simulation results acquired by Thiessen Polygon tool embedded in ICM were less different from the manual method based on a large amount of data and experience. Moreover, Thiessen Polygon was easy to divide the low-lying city plain area, which could save time and require less personnel experience. Therefore, Thiessen Polygon was suitable for the division of the subset water area in this project. The authors have added the explanation in Line 140, Page 4.

at line 106. you mentioned rainfall duration was 2 hours, however, at line 125 rainfall lasted for 2040 mins,

Response: Thanks for your carefulness. The authors are sorry for the misunderstanding caused by the inaccurate description. The 2 hours rainfall duration was the designed rainfall duration simulated by Chicago rainfall model, while the 2040 min is the actual rainfall duration we used to check the hydraulic model. The authors have modified the description in Line 126, Page 4.

153. Table 2. SS average value C is significantly higher as compared to other readings of SS. Why? you do not explain

Response: Thanks for your carefulness. The SS average value C in table 2 was the average of three measurements at the test time of 14:00. The reason why the value C is significantly higher in comparison with other SS is that the problem of the combined rainwater in this area is very serious, and the effluent is greatly affected by the surrounding mixed sewage. Therefore, due to the existence of restaurants and car washes around the test site, there will be a large amount of wastewater flowing into the pipe from 12:00 to 2:00, resulting in the obvious increase of SS value. The authors have added this reason in Line 172, Page 5.

line 161, 162. please describe the models you have mentioned, better idea is to include them in literature review so you can site them in methodology.

Response: Thanks for the comment. Both the Desbordes Model and the ackers white model were optional modules in the Infoworks ICM Model. Among them, the Desbordes Model was used to simulate the surface scouring process, and the Ackers White model was used to simulate the cumulative scouring process of pollutants.

line. 165. adjustment of catchment area? As I know you already defined your catchment boundary. how and why you need to adjust your catchment boundary.

Response: Thanks for your carefulness. The initial definition of the catchment boundary was defined by the simulated rainfall, while the actual catchment area was a little different from the simulated one. Thus, the authors obtained the range of the parameters through the previous investigation, and adjusted and optimized the parameters on the principle of minimum difference between simulated and measured water point distribution and water depth.

line 176. you refer to Figure 3. In text you have mentioned the time in minutes, however, the graph has been designed in hours, it’s difficult to correlate, please improve the figures and be consistent with units used in text and in figures.

Response: Thanks for your valuable comment. Sorry for our carelessness. The authors have modified each unit in figures to make sure they are consistent with that used in text.

line 241. equation is not explained and cited

Response: Thanks for your carefulness. The equation is a control mode for overflow pollution with the combination of water quality and flow, which is based on the years of work experience of the authors. This equation can not only ensure that the overflow sewage pollutants in rainy days meet the required discharge standard, but also provide the basis for the real-time online scheduling. The authors have added the explanation in Line 272, Page 10.

line 305. Conclusion is not enough, requires further improvements.

Response: Thanks for the valuable comment. The authors have improved the conclusion, and the changes are highlighted in red font in the part of the conclusion.

line 305. RTC?

Response: Thanks for the comment. Sorry for our carelessness. “RTC” is the abbreviation of “real-time control”. The authors have added the complete name where the word first appeared, and it can be found in Line 259, Page 10.

Reviewer 2 Report

The topic is of great interest today. However, the paper refers to a particular case, and the authors should reflect (possibly in the conclusions) on the possibility and manner of generalizing to other cases the approach that they propose.

It may be accepted that the strategic parameters considered are only COD and NH4, but this should be duly justified by the authors in the paper.

It should be noted that combined sewage systems have emerged in some countries (in Europe, for example, in the 19th century) due to the utilization of stormwater networks to drain domestic sewage, when this need begin to be recognized for public health reasons. A brief review of the introduction is suggested to improve the historical approach to the problem.

Author Response

The topic is of great interest today. However, the paper refers to a particular case, and the authors should reflect (possibly in the conclusions) on the possibility and manner of generalizing to other cases the approach that they propose.

Response: Thank you for your valuable comment. This intercepting strategy is based on the actual situation of the study area, and also has considered the application method. As for the control of the combined sewage pollution in different areas, this paper can provide the research method for reference. Therefore, through the research idea proposed in this paper, it is possible to apply this strategy to other cases. For example, this strategy can provide a control mode for the overflow pollution existed in different areas, which can not only ensure that the overflow sewage pollutants in rainy days meet the required discharge standard, but also provide the basis for the real-time online scheduling. Besides, the construction and application of the model proposed in this paper can also be referred in the process of other interception strategy. Furthermore, the RTC (real-time control) technology based on water quality interception and the method used for the optimization of interception strategy have a useful reference value for the optimization of application in different cases. The authors have added the explanation in Line 351, Page 13.

It may be accepted that the strategic parameters considered are only COD and NH4, but this should be duly justified by the authors in the paper.

Response: Thank you very much for your comment. According to the comparison of various interception methods presented in Table 8, it can be found that the intercepting effect based on SS and TP was the best. However, the construction cost of this method would be expensive due to the large flow of the intercepting sewage. In addition, the low pollution concentration of the interception based on COD in the later period would lead to no interception in the later stage, and the earlier period of the interception based on NH4-N was too short. Thus, considering that the correlation between SS, TP and COD was above 0.9, COD + NH4-N can be combined to intercept the flow from the perspective of the intercepting control. The authors have justified the reason in Line 306, Page 12.

It should be noted that combined sewage systems have emerged in some countries (in Europe, for example, in the 19th century) due to the utilization of stormwater networks to drain domestic sewage, when this need begin to be recognized for public health reasons. A brief review of the introduction is suggested to improve the historical approach to the problem.

Response: Thank you very much for your comment. The authors have improved the introduction in term of the problem you mentioned above, and the changes are highlighted in red font in Line 32, Page 1 and Line 67, Page 1.

Water EISSN 2073-4441 Published by MDPI AG, Basel, Switzerland RSS E-Mail Table of Contents Alert
Back to Top